# Evidence for the Hydration of Some Organic Compounds during Reverse-Phase HPLC Analysis

**DOI:** 10.3390/molecules28020734

**Published:** 2023-01-11

**Authors:** Igor G. Zenkevich, Abdennour Derouiche, Daria A. Nikitina

**Affiliations:** Institute for Chemistry, St. Petersburg State University, 198504 St. Petersburg, Russia

**Keywords:** reverse-phase HPLC, hydration of analytes, recurrent approximation of retention times, retention indices, dependence of indices on the concentration of an organic modifier in an eluent

## Abstract

Some polar analytes (X) can reversibly form hydrates in water-containing eluents under the conditions of reversed-phase HPLC analysis, X + H_2_O ⇄ X × H_2_O. One of the methods to detect their formation is the recurrent approximation of the net retention times of such analytes, *t*_R_(*C* + Δ*C*) = *at*_R_(*C*) + *b*, where Δ*C* = const is the constant step in the variation of the organic modifier content of an eluent. These dependencies are linear if hydrates are not formed, but in the case of hydrate formation, they deviate from linearity under high water content. It has been shown that UV spectroscopic parameters, namely, relative optical densities: *A*_rel_ = *A*(λ_1_)/*A*(λ_2_), depend on eluent composition for some organic compounds, but their variations cannot be used as indicators for hydrate formation. The coefficients that characterize the dependence of the analyte retention indices on the organic component concentration of an eluent, *d*RI/*dC*, appeared to be the most informative additional criterion for hydration. The values of these coefficients for most polar analytes are largely negative (*d*RI/*dC* < 0), whereas, for nonpolar compounds, they are largely positive (*d*RI/*dC* > 0).

## 1. Introduction

The principal advantage of high-performance liquid chromatography (HPLC) compared to gas chromatography is its applicability to nonvolatile and thermally unstable analytes [1]. At the same time, the main disadvantage of reversed-phase (RP) HPLC is the risk of the hydrolysis of some analytes due to the presence of water in the eluent. The “intermediate option”, which is rarely taken into account, is the reversible formation of the hydrates of some analytes during their chromatographic separation in water-containing eluents.

The formation of hydrates is the typical property of numerous inorganic compounds [2]. Most such hydrates are stable and can be isolated in a solid state. However, unexpectedly, many organic compounds (X) also form hydrated forms, preferably monohydrates. Instead of the expression “formation of hydrates”, the following equilibrium seems to be more rigorous:(1)$$X+H_{2}O⇄X\times H_{2}O$$

The probability of hydrate formation is determined by the constant of hydration, *K*_hydr_:*K*_hydr_ = [X × H_2_O]/{[X] × [H_2_O]}(2)

If *K*_hydr_ << 1, the formation of hydrates in an aqueous media can be neglected, but the inequality *K*_hydr_ >> 1 corresponds to relatively stable hydrates. Some of them can be isolated so that their physicochemical properties can be experimentally determined. CAS numbers are assigned to numerous hydrates, both stable and unstable. Several examples of hydrates of both kinds are presented in Table 1. Some organic compounds form stable covalent hydrates (e.g., trifluoroacetaldehyde, hexafluoroacetone, ninhydrin, etc.).

The information on hydrates in this table is taken both from original publications and (mostly) from the webpages of chemical companies (more detailed collections of the data for the hydrates are presented in [3,4]).

Because hydrates are compounds that are definitely more polar than the anhydrous forms of organic compounds, their formation may account for some anomalies of their retention in RP HPLC, depending on the ratio between water and the organic modifier in an eluent. If *K*_hydr_ << 1, the eluent contains solely the nonhydrated form of the analyte and there should be no anomalies of its retention under any eluent composition. On the other hand, if *K*_hydr_ >> 1, we can assume the predominance of the hydrated form of the analyte with which no transformations take place with variation in the eluent composition; hence, no retention anomalies are observed as well. The most interesting case is the comparable content of the nonhydrated and hydrated forms of analytes in an eluent (both forms coexist together), i.e., when *K*_hydr_ ≈ 1. In this case, the variations of the ratio of the organic and aqueous phases in an eluent should strongly influence the position of the equilibrium (2) and the ratio of the nonhydrated and hydrated forms, causing unpredictable variations in the retention parameters of such analytes. Numerous equations for approximation of the dependencies of the retention times on the content of organic modifiers have been proposed (see, e.g., [5,6]). It is important that most of them become inapplicable if the analytes reversibly form hydrates in an eluent. Actually, hydrate formation is the chemical transformation of an analyte during chromatographic separation depending on the organic modifier concentration.

Thus, the problem of the HPLC detection of reversibly formed hydrates is that the retention times correspond not to the sole structures but to at least two of the different forms of the analytes in variable proportions, depending on the eluent composition. Detecting relatively small anomalies in the *t*_R_-values against the background of their significant variations due to the dependence *t*_R_ = *f*(*C*) seems to be rather difficult. Let us briefly discuss the possible effects of hydrate formation using easily perceived examples.

The unusual anomalies of chromatographic retention caused by the formation of hydrates were revealed for the first time for several complex polyfunctional synthetic antitumor drugs produced by Biokad JSC (St. Petersburg, Russia) [7]. Because the direct presentation of the *t*_R_ values as a function of the concentration of the organic modifier in an eluent (*C*) does not reveal most of the anomalies, the so-called recurrent representation of the retention times was used:*t*_R_(*C* + Δ*C*) = *at*_R_(*C*) + *b*, (3)
where Δ*C* is the constant increment in the variations of the organic modifier concentration in an eluent, and the coefficients *a* and *b* are calculated by the least squares method (LSM).

A short description of the properties of the recurrent relations is discussed below (Section 3.1). Here, it seems important to compare the plots of the recurrent dependencies (3) for three drugs with the trivial names gefitinib (Figure 1, structure I), pazopanib (II), and imatinib (III):

All the plots are based on the raw retention times within acetonitrile concentration ranges of 35–65% *v*/*v* (compounds I and III) and 20–50% *v*/*v* (compound II), varied with 5% steps (seven experimental *t*_R_ values for each analyte). This gives six points for the recurrent dependencies, which lie on the same straight line for gefitinib (I) (correlation coefficient *R* = 0.9996). On the contrary, the corresponding plot for pazopanib (II) has a linear section (four “left” points) with *R* = 0.9998 but two “right” points that deviate (down) from the regression line. The right parts of all plots (largest argument values) correspond to the eluents that contain the largest amounts of water when the equilibria (1) are shifted toward the formation of more hydrophilic hydrates. Other deviations in the regression data approximations can be observed for imatinib (III): the four right points fall on a straight line (*R* = 0.9998), whereas the two left points (for the eluents with the highest acetonitrile content) deviate down from the regression line. This means that the recurrent approximation is a sensitive tool for revealing “fine” anomalies in retention data.

This information on hydrate formation seems to be rather important for different analytical applications. For example, the values of the so-called hydrophobicity factor (log*P*) are considered to represent the valuable characteristics of organic compounds, including drugs. Different kinds of software (e.g., ACD, ChemAxon, etc.) are recommended for the theoretical evaluation of these parameters. However, all such calculations can be made for nonhydrated molecules. If the target analyte forms a hydrate, the precalculated log*P* values are very different from the experimental values.

Because the detection of hydrate formation in chromatographic eluents appeared to be a difficult task (in particular), and the possibilities of the recurrent approximation of the retention data in RP HPLC require additional characterization (in general), we consider these problems in our paper.

## 2. Results

### 2.1. Measuring the Retention Times of Selected Analytes

Up to now, revealing the dependencies of the retention parameters (*t*_R_) on eluent composition (usually on the content of the organic modifier, *C*) remains the main trick in the HP HPLC characterization of various organic compounds on different sorbents (see, e.g., [8,9,10,11]). A few dozen different equations have been proposed for approximating this dependence *t*_R_(*C*) [6]. Our task was to characterize the features of those compounds forming the hydrates in eluents. For this purpose, we have selected about 30 model compounds for the analyses with the methanol–water eluents and about 20 compounds for the analyses with the acetonitrile–water eluents. The *t*_R_ measurement is a standard procedure that does not require special detailed description. It is unnecessary to consider all the numerical *t*_R_(*C*) data. The retention times of the selected analytes were measured within the ranges 50–85% vol. methanol and 35–70% vol. acetonitrile, with 5% concentration steps.

The important detail of the selection of the model analytes was to avoid the coincidence of their p*K*_a_ values with the pH of the eluents. For the acetonitrile-containing eluents without acidic or salt constituents, the reference pH value was approximately 5.6, whereas for the eluent 1:1 *v*/*v* methanol + 0.1% trifluoroacetic acid, the pH was 2.7. The p*K*_a_ values of some of the analytes are as follows: 1H-benzotriazole 8.5 ± 0.1, phthalimide: 8.2 or 10.2, 1-phenylpyrazolidin-3-one: 7.5 and 9.5, diethyl-*m*-toluamide: −1.37, all N-substituted *p*-toluenesulfonamides: 11.2 ± 0.8 (average value for 10 compounds), sulfamethoxazole: 5.7 ± 0.3, sulfamerazine: 7.0, and *p*-toluilic acid: 4.4. Due to the p*K*_a_ value, the latter acid was excluded from further consideration. The most “suspicious” value in the above series is 5.7 ± 0.3 for sulfamethoxazole (close to the 5.6 pH of the acetonitrile-containing eluents), but in the aqueous solution, this compound exists as hydrate, with a different p*K*_a_ value.

Even small uncontrolled variations in the eluent flow rate can affect the experimental results [12,13]. This is manifested to the greatest extent in the water–methanol eluents because the viscosity of the CH_3_OH–water mixtures is maximized under an approximate 40 vol.% methanol content. If the HPLC pump(s) does not provide the fixed eluent flow rate under increasing eluent viscosity, this may lead to unpredictable distortions of the *t*_R_-values. This is why the use of one of the available HPLC instruments to us was rejected [14].

### 2.2. Calculation of Retention Indices

As all the analyses of selected compounds were carried under isocratic conditions, all of them were characterized by logarithmic (Kovats) retention indices [15]:RI_x_ = RI_n_ + (RI_n+1_ − RI_n_) × [log(*t*_R,x_′) − log(*t*_R,n_′)]/[log(*t*_R,n+1_′) − log(*t*_R,n_′)](4)
where *t*_R,x_, *t*_R,n_, and *t*_R,n+1_ are the net retention times of the target analyte and the two reference compounds eluted immediately before and immediately after (*n*-alkyl phenyl ketones), and RI_x_, RI_n_, and RI_n+1_ are their retention indices and the prime means conversion of net retention times to the adjusted retention times, *t*_R_′ = *t*_R_ − *t*_0_, where *t*_0_ is the retention time of the theoretically unabsorbed component (“dead time”).

The required *t*_0_-values were calculated using the *t*_R_ values for the three serial homologs of the *n*-alkyl phenyl ketones using the Peterson and Hirsch relationship [16]:*t*_0_ = (*t*_R,1_*t*_R,3_ − *t*_R,2_^2^)/(*t*_R,1_ + *t*_R,3_ − 2*t*_R,2_) (5)

Relation (4) is equivalent to the following linear dependence (coefficients *a* and *b* are calculated by LSM):RI_x_ = *a*log(*t*_R,x_′) + *b*
(6)

This means that the calculation of the retention indices is possible not only by interpolation (“between” reference compounds) but, in some cases, by extrapolation (out of the range of the retention times of the reference compounds).

The retention indices of some of the organic compounds determined using methanol as the organic component of an eluent are listed in Table 2; the data for the acetonitrile-containing eluents are presented in Table 3.

The symbol *N*{H} in Table 2 and below means the total number of the so-called active hydrogen atoms in a molecule (the number of atoms capable of exchanging with the hydrogen atoms of a solvent). The intra- and interday reproducibility of the RI values in these tables is approximately 1–3 index units (i.u.).

### 2.3. Evaluation of the Relative Optical Densities

The detection of the hydrates of the analytes (X×H_2_O) formed in an eluent can be achieved, at least theoretically, by recording and interpreting the changes in their UV spectra. However, the registration of the absolute UV spectroscopic parameters in HPLC is not reliable enough; hence, the determination of the so-called relative optical densities (*A*_rel_) seems to be preferable:*A*_rel_ = *A*(λ_1_)/*A*(λ_2_) ≈ *S*(λ_1_)/*S*(λ_2_) (7)
where *S*(λ_1_) and *S*(λ_2_) are the areas of the same chromatographic peak at different wavelengths.

The relative optical densities were recommended as an additional criterion for the identification of the analytes using RP HPLC in combination with the chromatographic parameters [17,18,19], including the level of the so-called group identification (attribution to the corresponding homologous series with the same chromophores). Table 4 contains the *A*_rel_ values for some of the organic compounds measured with the methanol–water eluents (the range of the methanol content is 50–85% *v*/*v*), and Table 5 contains the analogous data for the acetonitrile–water eluents (55–70% *v*/*v* acetonitrile content).

Both tables contain examples of compounds with both ascending and descending dependencies *A*_rel_(*C*), as well as with almost no clearly pronounced dependencies. For instance, the aromatic hydrocarbons (toluene, *o*-xylene) in the methanol–water eluents demonstrate the ascending dependence *A*_rel_(*C*), while in the acetonitrile–water eluents, it slightly descends. The reference compounds in RP HPLC, *n*-alkyl phenyl ketones, are characterized by *dA*_rel_/*dC* < 0 in all eluents. The most interesting objects, the N-substituted *p*-toluenesulfonamides, demonstrate practically no dependence regarding their relative optical densities on eluent composition.

The joint consideration of the data in Table 4 and Table 5 allows for the following conclusions: (1) the *A*_rel_ values for the analytes in the methanol–water and acetonitrile–water eluents are not usually equal to each other; (2) in some cases, these values depend on the eluent composition, and (3) the variations of these parameters, depending on the organic modifier concentration, are not directly related to hydrate formation. Despite the negative character of this conclusion, it seems rather important because it prevents further attempts to use spectral parameters for detecting the formation of hydrates.

## 3. Discussion

### 3.1. Recurrent Approximation of Chemical Variables: Important Features

The simple first-order linear recurrent regressions can be applied to the monotonic functions (*A*) of the integer (*n*) (Equation (8)) or the equidistant values of the argument (Δ*x*) (Equation (9)). The first kind of recurrence is applicable to the approximation of the various physicochemical properties of the homologs (functions of the number of carbon atoms in molecules), with the number of carbon atoms being an argument by definition [20]. The second kind of recurrence allows for their application to the functions of the temperature or pressure of chemical systems, as well as the concentrations of their constituents. In the latter case, the steps of the variation of the arguments, Δ*x*, should be fixed:*A*(*n* + Δ*n*) = *aA*(*n*) + *b*
(8)
*B*(*x* + Δ*x*) = *aB*(*x*) + *b*, Δ*x* = const (9)

Specifically, the latter type of relationship can be used in the approximation of the chromatographic retention parameters as functions of temperature (gas chromatography) or of the organic modifier content of eluent in RP HPLC [21].

Recurrent relationships have several unusual mathematical properties. First, their mathematical equivalent (e.g., for Equation (8)) is the polynomial of the variable degree:*A*(*n*) = *ka*^n^ + *b*(*a*^n^ − 1)/(*a* − 1) (10)

Hence, recurrence relationships unite the properties of the arithmetic (at *a* ≡ 1 and *b* ≠ 0) and geometric (at 0 < *a* ≠ 1 and *b* ≡ 0) progressions. This fact accounts for their unique approximating “ability”, especially for the various properties of the homologs within a homologous series because the number of carbon atoms in the molecule cannot be a noninteger argument by definition. Examples of the applicability of these recurrences to the equidistant values of pressure, temperature, or the concentrations of the constituents are the dependencies of *t*_R_ on the temperature in gas chromatography and on the organic modifier content of an eluent in RP HPLC [21]. It is noteworthy that using the recurrent relationships in both gas chromatography and HPLC does not require the preliminary determination or calculation of the so-called “dead” time (*t*_0_). Another feature that seems to be important for plotting the recurrent dependencies is that the values of the arguments are not represented in such plots; every point is fixed by the two “neighboring” values of the functions.

When applied to the retention parameters in RP HPLC, recurrence relationships (9) are most often characterized by the correlation coefficients *R* > 0.999 for those analytes that show no anomalies in their chemical nature (e.g., not involved in prototropic equilibria and form no hydrates or tautomers). However, if two (or more) forms of analytes are present in an eluent (e.g., when *K*_hydr_ ≈ 1), deviations from the linearity of the recurrent dependencies (3) can be expected; this is due to the fact that the approximation “ability” of the recurrence is significant but not infinite; any changes in analyte speciation lead to distortions in the linearity of the recurrent dependencies.

Thus, the detection of the reversible formation of the hydrates of organic compounds in aqueous media (including HPLC eluents) should be based on the detailed consideration of the dependencies, *t*_R_(*C*). The first approach seems to be just revealing the deviations of the recurrent approximation of the net retention times from the linearity under high-water-content eluent. The second approach, which is discussed in this manuscript, considers the features of the retention indices of the analytes in HPLC.

### 3.2. Numerical Modeling of the Anomalies in the Recurrent Approximation of Retention Times

The application of recurrent approximation for revealing the formation of the hydrates of analytes can be illustrated by the following numerical example.

If an analyte (X) forms a hydrate (X×H_2_O) in an eluent, then its retention time can be expressed (very roughly, without considering the details) as an arithmetic mean of the retention times of the nonhydrated and hydrated forms:*t*_R_ ≈ [*t*_R_(X) + *t*_R_(X×H_2_O)]/2 (11)

It is logical to believe that the hydrated form of an analyte is more hydrophilic than the nonhydrated form; hence, *t*_R_(X×H_2_O) < *t*_R_(X), or *t*_R_(X×H_2_O)(*C* − *y*) ≈ *t*_R_(X)(*C*).

In order to simplify our numerical model, let us assume that the retention factor (*k*) is inversely proportional to the volume fraction of the organic component of an eluent (*x*). Such prerequisites correspond to the Row model [22]:1/*k* = *ax* + *b*(12)

Moreover, let us postulate that *a* = 1 and *b* = 0, using the net retention times instead of the *k*-values, which gives *t*_R_~1/*x*. Let us also imagine that the content of the organic modifier in an eluent varies from 0.4 to 1.0 with a step of 0.05. This gives the following set of *t*_R_-values:
**0.4 ≤ *x* ≤ 1.0****0.4****0.45****0.5****0.55****0.6****0.65****0.7****0.75****0.8****0.85****0.9****0.95****1.0***t*_R_2.502.222.001.821.671.541.431.341.251.181.111.051.00

The plot of this dependence *t*_R_(*x*) is the plot of a hyperbolic function (Figure 2a).

The recurrent approximation (Equation (3)) of the same data set with Δ*C* = 5% steps gives a linear regression with the following parameters: *a* = 0.847 ± 0.006, *b* = 0.118 ± 0.010, *R* = 0.9998, and *S*_0_ = 0.009; the plot of this dependence is shown in Figure 2b. It looks typical for analytes forming no hydrates in eluents (having no anomalies of chromatographic retention).

Let us assume that analyte X forms more hydrophilic hydrate X×H_2_O and that the retention time of this hydrate approximately corresponds to that of the parent compound X at the higher content of the organic modifier, *t*_R_(X×H_2_O)(*C*) ≈ *t*_R_(X)(*C* + *y*). If we accept *y* = 0.2, we obtain the following set of numerical estimations and, finally, the target retention times, *t*_R_* (the last line below):
0.2 ≤ *x* ≤ 0.80.20.250.30.350.40.450.50.550.60.650.70.750.8*t*_R_(non-hydrate)5.004.003.332.862.502.222.001.821.671.541.431.341.250.4 ≤ *x* ≤ 1.00.40.450.50.550.60.650.70.750.80.850.90.951.0*t*_R_(hydrate)2.502.222.001.821.671.541.431.341.251.181.111.051.00*t*_R_* = [*t*_R_ + 
*t*_R_(hydrate)]/23.753.112.662.342.081.881.721.581.461.361.371.191.12

Surprisingly, the recurrent approximation of the set of *t*_R_*-values (the sum of two hyperbolic functions) in comparison with the plot in Figure 2b visually demonstrates the detected deviations from linearity (Figure 3) only in the area of the large *t*_R_*-values, corresponding to the high water content of an eluent.

Such features of the recurrent approximation plots are typical of analytes forming hydrates in eluents.

### 3.3. Revealing Those Compounds That Are Reversibly Forming Hydrates

Hydrate formation can be readily confirmed for solid substances (both inorganic and organic). This can be carried out using differential scanning thermogravimetry or even “classical” elemental analysis. The detection of the unstable hydrates of organic compounds in solutions (when their isolation is impossible) is much more difficult. In some cases, hydrate formation can be inferred from the appearance of new bands in the IR and UV spectra. Mass spectrometric methods provide no information on the formation of hydrates in solutions.

As mentioned above, some polyfunctional synthetic drugs were the first examples of the application of recurrent approximation of net retention times in RP HPLC, revealing reversible hydrate formation [7]. Most of these drugs contain polar functional groups, including amides or sulfonamides. A literature search showed that the formation of hydrates both in the solid state and in aqueous solutions is the typical chemical property of such compounds [23,24,25,26,27,28,29]. Therefore, we have specially synthesized a series of monofunctional N-substituted *p*-toluenesulfonamides [CH_3_-C_6_H_4_-SO_2_-NRR′ (R, R′ = H, -CH_2_CH=CH_2_; (I); -(C_2_H_5_)_2_ (II); H, *tert*-C_4_H_9_ (III); H, -C_6_H_5_ (IV); H, -CH_2_C_6_H_5_ (V); H, -C_6_H_13_ (VI)] as appropriate model objects that can form hydrates in an eluent.

A comparison of the structures of gefitinib (I), pazopanib (II), and imatinib (III) (Figure 1) shows that the structure (I) contains no sulfonamide or amide groups, and it exhibits no anomalies in the recurrent approximation of the retention times (Figure 1a). Structure (II) demonstrates the deviations from linearity in the right part of the plot, corresponding to long retention times for the eluents with high water content (Figure 1b). However, structure (III), on the contrary, exhibits a recurrent anomaly in the area of the small retention times corresponding to the low water content of the eluent. This example deserves special comment because this anomaly is obviously not related to hydration.

Imatinib is a complex polyfunctional compound. Its molecule contains at least four possible nonconjugated sites for proton location. Hence, it is characterized by at least four p*K*_a_ values. They are (both experimental and precalculated (ChemAxon) values) 8.1–8.3, 3.7–4.0, 2.5, and 1.5. For our further consideration, it is important that some of these precalculated values may differ from the experimental data by 0.5–1.0.

Thus, we have a compound with one p*K*_a_ value of approximately 2.5, which is close to the pH of the eluent, 2.7–2.9. This means that two forms of this analyte (nonprotonated and protonated) exist in equilibrium in the solution:X + H^+^ ⇄ XH^+^
(13)

Moreover, increasing the concentration of the organic component of a solvent usually leads to an increase in the p*K*_a_ values of the dissolved compounds, which can be illustrated by the dependence of the p*K*_a_ values of 2-hydroxy-4-methyl-1,3,2-dioxaphospholane 2-oxide (trivial name propylene hydrogen phosphate) on ethanol concentration in aqueous solutions [30]:
***C*(C_2_H_5_OH)****0****50****80****95**p*K*_a_1.752.853.214.54

Apparently, this specific effect is responsible for the deviations in the recurrent approximations from linearity, as is the case for Imatinib (III). With an increase in the acetonitrile content of the eluent, the p*K*_a_ of this compound (about 2.5 in water solutions) moves closer to the pH ≈ 2.7–2.9 of the eluent. The coexistence of two forms of the analyte makes the recurrent approximation of its retention time nonlinear (see anomalies in Figure 1c).

Important information on the applicability of recurrent approximation is provided by comparing the sets of retention times of the same compounds on HPLC columns of different polarities. Figure 4a (five points) presents the retention times (45–70% *v*/*v* CH_3_CN content) of N-hexyl-*p*-toluenesulfonamide measured with a nonpolar EC-C18 column using acetonitrile–water eluents. Four points (excluding the right point) fall within a straight line according to the correlation coefficient *R* = 0.9999. The plot in Figure 4b presents the retention times (50–85% *v*/*v* CH_3_CN content) of the same sulfonamide measured with a slightly more polar EC-CN column. The six points, without the right point, correspond to a straight line with *R* = 0.9999. In both cases, the right points visibly deviate from the linearity. Hence, the effect observed has no relation to the column polarity and is determined only by the variations in eluent composition.

The plot in Figure 4c demonstrates the recurrent approximation of the retention times measured with a nonpolar C18 column using the methanol–water eluents (55–85% *v*/*v* CH_3_OH). In contrast to Figure 4a,b, the deviation of the right point from linearity is negligible (correlation coefficient for five points without the right point is 0.9998 and is *R* = 0.9995 with this point). The lower “sensitivity” of the methanol–water eluents to the formation of hydrates was discussed in [4]. Methanol itself forms rather stable monohydrates (the free energy of methanol hydration was estimated experimentally as (−5.1) kcal mol^−1^ [31]), which can effectively prevent the formation of hydrates in other compounds.

### 3.4. Retention Indices in Reversed-Phase HPLC: An Alternative Way to Suppose the Formation of Hydrates

The concept of retention indices (RI, Equation (4)) in reversed-phase HPLC appeared to be somewhat less popular than in gas chromatography, despite various applications [15,32,33]. This is caused by a dependence on a larger number of parameters (than in GC) (first, by the influence of different additives in the eluents) and, in general, by the narrower ranges in variation. Another reason is that the dependencies of the retention parameters of the analytes on the concentrations of organic solvents in eluents are complex [6].

The RI values for selected compounds are presented in Table 2 and Table 3. Apparently, these data provide no information on the reversible formation of hydrates in an eluent. Hence, RI values should be transformed into more informative parameters.

One of the important properties of the GC retention indices is their temperature dependence, RI = *f*(*T*). Usually, the presentation of this function is limited to the first term of its expansion in a Taylor series, β = *d*RI/*dT* [15]:RI(*T*) = RI(*T*_0_) + *d*RI/*dT* (*T* − *T*_0_), (14)
where *T*_0_ is any temperature conventionally chosen as a standard for data presentation (usually 0 or 100 °C).

This secondary parameter based on the GC retention indices depends on the differences in the topological characteristics of analytes and reference *n*-alkanes. For the majority of organic compounds, the coefficients, β, obey inequality β > 0. The *d*RI/*dT* values increase with an increasing number of branches for the molecular carbon skeleton, as well as with the number and size of the rings. Specifically, the large absolute *d*RI/*dT* values are responsible for the low interlaboratory reproducibility of GC retention indices [34].

The analog of the temperature dependence of the GC retention indices in RP HPLC is the dependence of the indices on the concentration of the organic solvent in an eluent, *d*RI/*dC* (Equation (15)). Unlike gas chromatography, the coefficients *d*RI/*dC* can be either greater or less than zero. Two dependencies RI = *f*(*C*) are plotted in Figure 5 for toluene (a, *d*RI/*dC* > 0) and for N-phenyl-*p*-toluenesulfonamide (b, *d*RI/*dC* < 0). In both cases, good linearity is observed (a, *R* = 0.9998; b, *R* = −0.998); the deviations from linearity for some analytes are caused by their tautomeric transformations or prototropic equilibria.
RI(*C*) = RI(*C*_0_) + *d*RI/*dC* (*C* − *C*_0_), (15)
where *C* is any concentration of an organic modifier chosen as a standard for data presentation (conventionally *C*_0_ = 0).

It should be noted that not the RI values and, specifically, the coefficients *d*RI/*dC* can be considered for additionally confirming the formation of the hydrates of analytes in an eluent. Comparing these coefficients (data are presented in Table 6) shows that the minimal values of *d*RI/*dC* belong to the most polar analytes, such as the N-substituted *p*-toluenesulfonamides, and the maximal values belong to less polar analytes, such as hydrocarbons (toluene, *o*-xylene) and their chloroderivatives (chlorobenzene). Table 6 presents the *d*RI/*dC* data for selected compounds, listed in increasing order and subdivided into three subgroups: low (*d*RI/*dC* ≤ −1.0), close to zero (−0.4 ≤ *d*RI/*dC* ≤ 0.3), and high (≥1.6). The first subgroup (nine most polar compounds) constitutes six sulfonamides with polar fragments –SO_2_–N<, one amide (–CO–N<), one cyclic hydrazide (–CO–NH–N<), and nitrophenol. The third subgroup includes only nonpolar compounds. Thus, we can conclude that the main factor that determines the sign and absolute values of the coefficients *d*RI/*dC* is the polarity of the analytes. The most negative values belong to the most polar sulfonamides, for which the probability of hydrate formation is maximal.

The easy hydration of sulfonamides and, to a lesser extent, of amides can be explained by the formation of two hydrogen bonds in the ring. This ring contains a S=O double bond (two π-electrons) and two pairs of *p*-electrons located at the N- and O-atoms (in total, sixπ- and *p*-electrons). In accordance with (4*n* + 2) Huckel’s rule, such systems exhibit pseudo-aromatic properties:



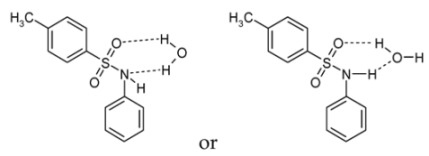



The set of compounds in the middle subgroup seems to be rather unusual. It contains four medium-polarity analytes (acetophenone, acetophenone hydrazone, nitrobenzene, and trimethylphenol) and four polar compounds: sulfamethoxazole (stable hydrate exists), ninhydrin (the same), 1H-benzotriazole, and phthalimide (formation of hydrates is rather probable). At the same time, the absolute values of the coefficients *d*RI/*dC* are not as large as those for the analytes of the first subgroup. It is interesting to note that, for acetonitrile-containing eluents, the *d*RI/*dC* values for sulfamethoxazole, 1H-benzotriazole, phthalimide, and trimethylphenol are less than -1.4, which corresponds to the compounds that are able to form hydrates. If the main reason for large negative *d*RI/*dC* values is the strong dependence of the equilibrium (of hydration equation (1)) on the content of the organic solvent in the eluent, then a lack of such dependence may be caused by the fact that the position of this equilibrium is independent of the solvent composition. In other words, the hydrate forms of some analytes from this subgroup exist under different compositions of the solvent. For example, ninhydrin (the parent structure contains no active hydrogen atoms) forms such a stable hydrate that it was characterized by p*K*_a_ 8.47, like typical organic acids [7]:



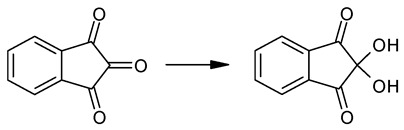



In order to finalize the consideration of the dependencies of the retention indices on the concentration of the organic constituents of an eluent, the following should be noted:-The *d*RI/*dC* coefficients for the same compounds are not equal to each other in methanol- and acetonitrile-containing eluents. Nevertheless, for the entire set of compounds, their values satisfactorily correlate with each other (correlation coefficient *R* is approximately 0.87);-The values for *d*RI/*dC* depend on the polarity of the organic compounds but show no correlation with hydrophobicity factors (log*P*), the number of active hydrogen atoms in a molecule, or the retention indices (RI);-For compounds of a different chemical nature, the values for *d*RI/*dC* are usually different. This means that if we need to improve the separation of two analytes in a different homologous series, we can slightly change the ratio of the organic and water components of the eluent. However, this recommendation may be ineffective if such a problem arises for compounds that are similar in nature (isomers or homologs).

## 4. Materials and Methods

### 4.1. Analytes, Reagents, and Solvents

The following compounds were used: toluene, *p*-xylene, chlorobenzene, nitrobenzene (all of reagent grade, for chromatography, Reakhim, Moscow, Russia), 1-phenylpyrazolidin-3-one (reagent grade, Reakhim, Moscow, Russia), 1H-benzotriazole (for photography, Reanal, Budapest, Hungary), acetophenone, propiophenone, butyrophenone (Sigma–Aldrich Rus, LLC, Russia), 2,3,5-trimethylphenol [Theodor Schuchardt, Munich, Germany (the sample from plant volatile compounds collection of Ph.D. S. Kozhin, Leningrad State University], 3-nitrophenol (indicator, British Drug Houses, Ltd., Great Britain), and *m*-toluilic acid diethylamide (DETA, insect repellent, TU (Technical Specification) 2386-077-00205357-2007). All the selected analytes were chosen so that their p*K*_a_ values did not coincide with the pH of eluents. Some synthetic antitumor drugs discussed in the text [gefitinib (I), pazopanib (II), and imatinib (III)] were produced by BIOCAD JSC (St. Petersburg, Russia) and are characterized in [7].

The series of N-alkylsubstituted *p*-toluenesulfonamides was synthesized by Ph.D. Tatiana A. Kornilova (St. Petersburg State University) from the corresponding amines and *p*-toluenesulfonyl chloride [35].
CH_3_–C_6_H_4_–SO_2_Cl + 2HNRR′ → CH_3_–C_6_H_4_–SO_2_–NRR′ + RR′NH·HCl
R, R′ = H, -CH_2_CH=CH_2_; (I); (C_2_H_5_)_2_ (II); H, *tert*-C_4_H_9_ (III); H, -C_6_H_5_ (IV); H, -CH_2_C_6_H_5_ (V); H, -C_6_H_13_ (VI).

The reaction mixtures were analyzed directly because excess amounts of amines and their salts do not hinder the UV detection of reaction products, which (except aniline) do not absorb in the near-UV region. The presence of certain amounts of *p*-toluenesulfonic acid (in the form of the anion) follows from the appearance of peaks in the region of the retention time of the non-sorbable component.

The stock solutions of all the analytes or reaction mixtures were prepared in 2-propanol (reagent grade, Kriokhrom, St. Petersburg, Russia) and were additionally diluted with an eluent for HPLC. To prepare eluents, we used deionized water (resistivity 18.2 MΩ cm) prepared using a Milli-Q device (Millipore, USA), acetonitrile (99.5%, HPLC-gradient grade, PanReac, Spain), and methanol (analytical grade, Kriokhrom, St. Petersburg, Russia). Some eluents contained 0.1% formic acid (98% analytical grade, PanReac, Spain) or 0.1% trifluoroacetic acid. Acetonitrile-containing eluents were degassed via filtration under vacuum and sonication in a 420 W Sapfir TTTs unit (Sapfir, Russia).

### 4.2. Conditions of HPLC Analysis

Chromatographic analyses of both individual analytes and reaction mixtures were performed in three regimes:

(A): Agilent 1260 Infinity liquid chromatograph with a diode-array detector (scanning range 220–340 nm) and an Infinity Lab Poroshell 120 EC-C18 column 50 mm long and 3.0 mm in diameter with a sorbent particle size of 2.7 μm in water–acetonitrile mobile phases in several isocratic modes with 5% concentration steps of the organic component at an eluent flow rate of 0.4 mL min^−1^ and a column temperature of 40 °C. For the analyses of the drugs, trifluoroacetic acid was added to the eluent to a 0.1% concentration; the pH of the eluent with 50% acetonitrile content was 2.7–2.9. All the model compounds were analyzed without any acidic or salt additives added to the eluent (pH of eluents about 5.6). Samples were injected using an SN G1329A autosampler; the sample volume was 5 μL.

(B): The same chromatograph (at the same scanning range) with Agilent Poroshell 120 EC-CN columns 100 mm long and 3.0 mm in diameter with a sorbent particle size of 2.7 μm in water–acetonitrile mobile phases in several isocratic modes with 5% concentration steps of the organic component at an eluent flow rate of 0.5 mL min^−1^ and column temperature of 40 °C. The samples were injected using an SN G1329A autosampler; the sample volume was 5 μL.

(C): Shimadzu LC-20 Prominence liquid chromatograph with a diode-array detector (scanning range 190–800 nm) and Phenomenex C18 columns 250 mm long and 4.6 mm i.d. with a sorbent particle size of 5 μm in water–methanol mobile phases with the addition of 0.1% formic acid (pH of aqueous solution was 5.6) in several isocratic modes with 5% concentration steps of the organic component at an eluent flow rate of 1.0 mL min^−1^ and column temperature of 30 °C. The samples were injected using a SIL-20A/AC autosampler; the sample volume was 20 μL.

All the samples for the analyses were prepared by dissolving individual compounds or reaction mixtures in the mobile phase. The number of replicate injections of each sample in all the regimes (A)–(C) was 2–3. The interinjection variations of the retention times of the target analytes in all the cases did not exceed 0.01–0.02 min. To determine the retention indices, a mixture of three reference *n*-alkyl phenyl ketones C_6_H_5_COC_n_H_2n+1_ with *n* = 1–3 was added to all the samples.

### 4.3. Data Processing

Chromatograms in regimes (A) and (B) were obtained, processed, and stored using the Mass Hunter software (Agilent Technologies, USA). The data were statistically processed using Excel software (Microsoft Office, 2010). Origin software (versions 4.1 and 8.1) was used for calculating the parameters of the recurrent dependencies and plotting all the dependencies. The logarithmic retention indices in the isocratic regimes were calculated using Excel software or manually (with calculators).

## 5. Conclusions

For some polar organic compounds, we can guess the reversible formation of their hydrates during reversed-phase HPLC separation, X + H_2_O ⇄ X×H_2_O. However, hydration confirmation seems to be a complex problem. The testing of the so-called relative optical densities, *A*_rel_ = *A*(λ_1_)/*A*(λ_2_), shows their dependence on the composition of eluents in some cases, but, in general, they exhibit inapplicability to the detection of hydrate formation.

One of the methods to detect the formation of hydrates seems to be the recurrent approximation of the net retention times of analytes, *t*_R_(*C* + Δ*C*) = *at*_R_(*C*) + *b*, where Δ*C* is the constant step in the variations of the organic modifier content of an eluent. In the case of hydrate formation, such dependencies deviate from linearity for large retention times, e.g., for eluents with high water content.

The coefficients that characterize the dependence of the retention indices on the concentration of the organic component in an eluent, *d*RI/*dC*, are suggested to represent an additional criterion for revealing the hydration of analytes during their reverse-phase HPLC analysis. The values of these coefficients for nonpolar compounds are largely positive (*d*RI/*dC* > 0), whereas, for most polar analytes, they are largely negative (*d*RI/*dC* < 0). The compounds of the latter type can, by themselves, form hydrates in HPLC eluents.

Is it possible to correct the anomalies caused by the formation of hydrates? The simplest possible way to not exclude these anomalies and minimize them is to replace acetonitrile in an eluent with methanol.

## Figures and Tables

**Figure 1 molecules-28-00734-f001:**
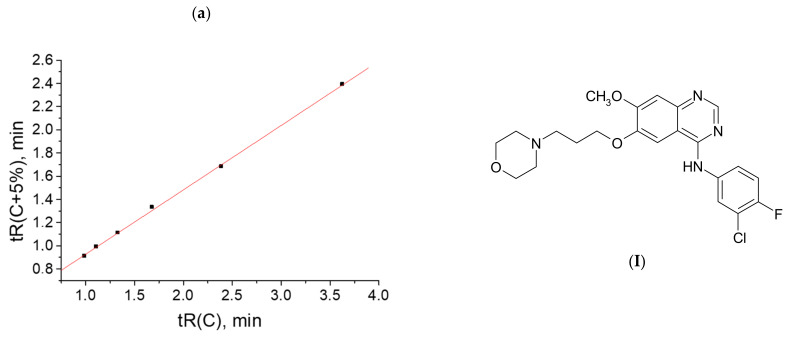
Examples of some of the typical features in the recurrent approximation plots of the retention times of (**a**) gefitinib (**I**), (**b**) pazopanib (**II**), and (**c**) imatinib (**III**); all data were obtained with acetonitrile–water eluents. See text for detailed comments.

**Figure 2 molecules-28-00734-f002:**
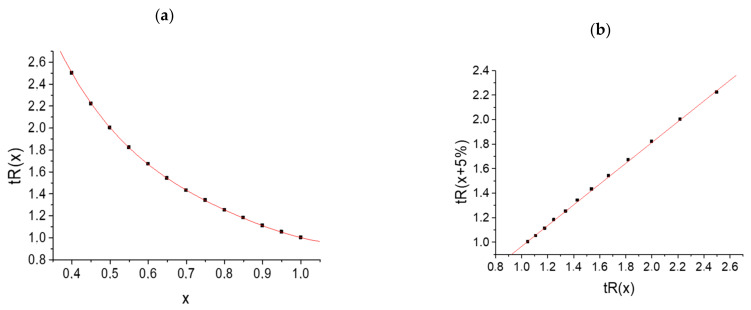
(**a**) Typical nonlinear dependence *t*_R_(*x*) (numerical modeling) and (**b**) linear recurrent approximation for the same set of data.

**Figure 3 molecules-28-00734-f003:**
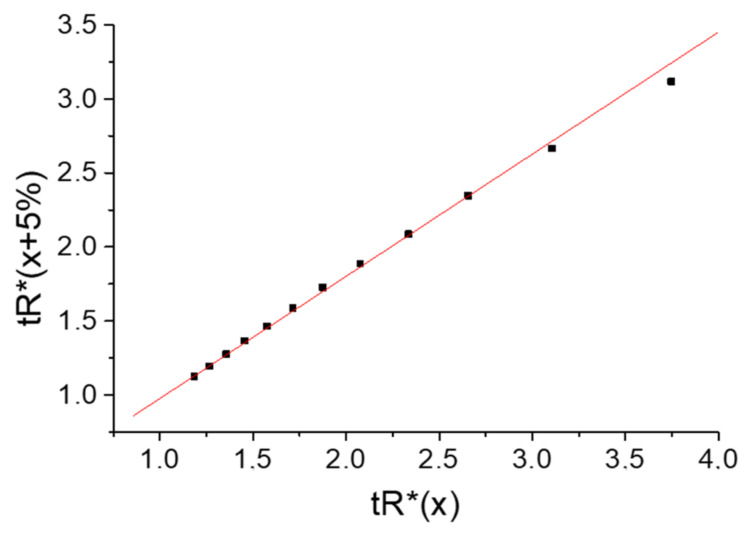
Plot of the recurrent approximation of the superposition of two nonlinear hyperbolical dependencies *t*_R_(*x*) (numerical modeling). The area of maximal *t*_R_*-values shows visible deviations from linearity.

**Figure 4 molecules-28-00734-f004:**
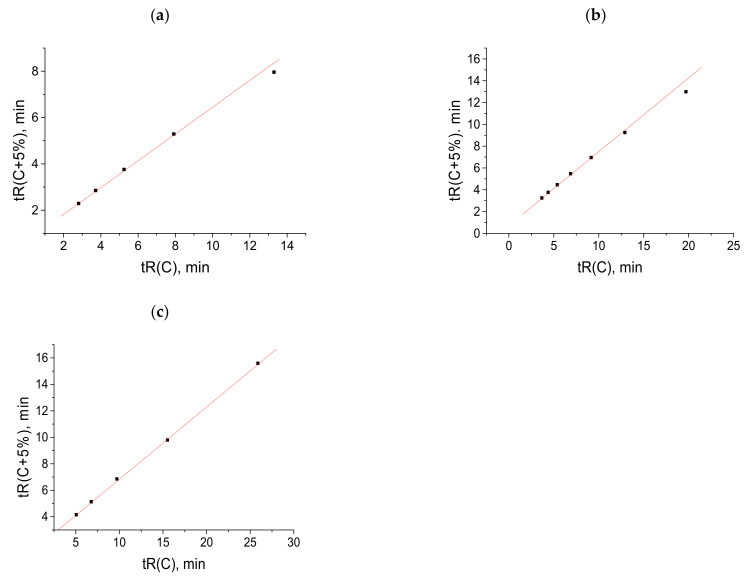
Recurrent approximation of the net retention times of N-hexyl-*p*-toluenesulfonamide, measured with (**a**) a 120 EC C18 column (eluent acetonitrile–water), (**b**) a 120 EC-CN column (the same eluent), and (**c**) a Phenomenex C18 column (eluent methanol–water).

**Figure 5 molecules-28-00734-f005:**
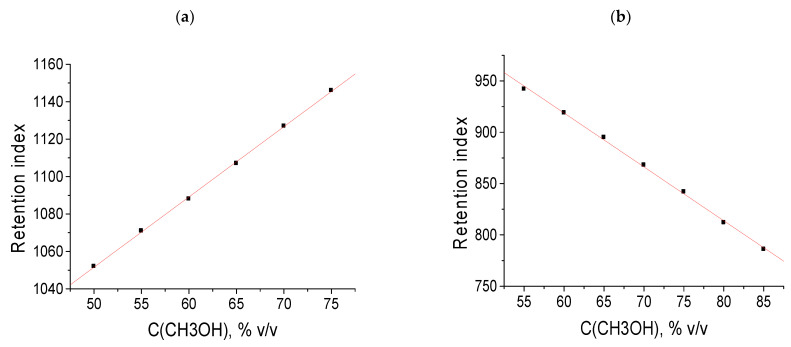
Two examples of the dependencies of the retention indices in RP HPLC on the concentration of methanol in an eluent: (**a**) toluene and (**b**) N-phenyl-*p*-toluenesulfonamide.

**Table 1 molecules-28-00734-t001:** Some examples of reference data for hydrates of organic compounds.

Compound	CAS No. (Anhydrous Form)	CAS No. (Hydrate)	Composition and Properties * (if Known)
Unstable hydrates			
Methanol	67-56-1	118240-86-1 151900-28-5	1:1
Acetonitrile	75-05-8	128870-13-3	1:1
Acetic acid	64-19-7	19215-29-3 99294-94-7	1:1, 1:2
Anthracene	120-12-7	188974-01-8	1:1
Stable noncovalent hydrates			
Ethylene diamine	107-15-3	6780-13-8	1:1; *T*_b_ 118; *n*_D_^20^ 1.448-1.451; *d*_4_^20^ 0.96
Citric acid	77-92-9	5949-29-1	1:1 **
Caffeine	58-08-2	5743-12-4	1:1 **
Benzene-1,2,3-tricarboxylic (hemimellitic) acid	569-51-7	732304-21-1 (mono); 36362-97-7 (di)	1:1; *T*_m_ 190-192

(*) Abbreviations: *T*_b_—normal boiling point, *T*_m_—melting point, *n*_D_^20^—index of refraction, *d*_4_^20^—relative density; (**) hydrates decompose below the melting point.

**Table 2 molecules-28-00734-t002:** Retention indices of some organic compounds, depending on the methanol content in the eluent.

Analyte	MW	*N*{H}	Methanol Content (% *v*/*v*)							
			50	55	60	65	70	75	80	85
Toluene	92	0	1052	1071	1088	1107	1127	1146	1176	-
*o*-Xylene	106	0	1150	1166	1186	1202	1230	1254	1292	-
Chlorobenzene	112	0	1046	1057	1067	1078	1090	1105	1127	1144
1H-Benzotriazol	119	1	688	686	684	684	684	680	681	-
Acetophenone	120	0	800	800	800	800	800	800	800	800
4-Methylbenzaldehyde	120	0	870	-	874	-	879	-	884	-
2-Hydroxybenzaldehyde	122	1	795	-	804	-	811	-	819	-
Nitrobenzene	123	0	847	849	854	856	860	857	852	-
Acetophenone hydrazone	134	2	-	738	737	738	734	737	736	734
4-Methylbenzaldehyde hydrazone	134	2	-	719	720	720	721	724	724	722
2,3,5-Trimethylphenol	136	1	852	854	858	862	862	864	860	-
2-Hydroxybenzaldehyde hydrazone	136	3		722	719	715	713	711	708	702
3-Nitrophenol	139	1	799	796	793	791	784	776	768	-
Phthalimide	147	1	693	692	691	689	690	686	686	-
4-Methylacetophenone hydrazone	148	2	-	784	799	803	830	-	-	-
Butyrophenone hydrazone	162	2	-	875	885	894	901	890	905	939
1-Phenylpyrazolidin-3-one	162	1	730	714	692	670	732	729	715	-
Ninhydrine (hydrate)	178	2	662	664	664	666	668	664	663	-
N-Allyl-*p*-toluenesulfonamide	211	1	852	838	823	808	792	772	756	732
N,N-Diethyl-*p*-toluenesulfonamide	227	0	978	964	950	936	920	903	885	862
N-*tert*-Butyl-*p*-toluenesulfonamide	227	1	968	952	935	918	898	876	852	824
N-Phenyl-*p*-toluenesulfonamide	247	1	963	942	918	895	869	842	813	782
N-Hexyl-*p*-toluenesulfonamide	255	1	1225	1205	1185	1165	1140	1110	1075	1029
N-Benzyl-*p*-toluenesulfonamide	261	1	1014	993	972	948	921	894	860	828

**Table 3 molecules-28-00734-t003:** Retention indices of some organic compounds depending on the acetonitrile content in the eluent.

Analyte	MW	Acetonitrile Content (% *v*/*v*)							
		35	40	45	50	55	60	65	70
Toluene	92	1028	1022	1024	1036	1036	1030	1048	-
*o*-Xylene	106	-	-	1119	1126	1122	1124	1151	-
Chlorobenzene	112	-	1035	1024	1036	1040	1036	1055	-
1H-Benzotriazol	119	694	668	652	654	644	630	654	-
Acetophenone	120	800	800	800	800	800	800	800	-
Nitrobenzene	123	863	864	858	865	860	845	851	-
2,3,5-Trimethylphenol	136	935	925	914	916	905	888	893	-
3-Nitrophenol	139	777	769	755	755	745	724		-
Phthalimide	147	711	692	679	685	676	666	686	-
1-Phenylpyrazolidin-3-one	162	697	671	654	654	642	-	-	-
Diethyl-*m*-toluamide	191	874	865	850	854	846	831	843	-
N-Allyl-*p*-toluenesulfonamide	211	879	870	861	848	837	827	814	816
N,N-Diethyl-*p*-toluenesulfonamide	227	-	1030	1021	1012	1006	997	987	974
N-*tert*-Butyl-*p*-toluenesulfonamide	227	972	962	951	939	930	921	914	903
N-Phenyl-*p*-toluenesulfonamide	247	1010	1005	970	948	928	904	883	-
Sulfamethoxazole (hydrate)	253	717	714	698	699	689	-	-	-
N-Hexyl-*p*-toluenesulfinamide	255	-	-	1208	1184	1174	1157	1139	1129
N-Benzyl-*p*-toluenesulfonamide	261	-	1025	1006	984	967	950	928	-
Sulfamerazine	264	694	668	652	651	638	621	-	-

**Table 4 molecules-28-00734-t004:** Relative optical densities *A*(254)/*A*(220) of some organic compounds, depending on the methanol content in the eluent.

Analyte	MW	Methanol Content (% *v*/*v*)							
		50	55	60	65	70	75	80	85
Toluene	92	0.14	0.21	0.24	0.26	0.30	0.34	0.37	-
*o*-Xylene	106	0.08	0.08	0.09	0.10	0.11	0.13	0.14	-
Chlorobenzene	112	0.032	0.031	0.031	0.031	0.031	0.031	0.032	0.028
1H-Benzotriazol	119	3.4	3.0	2.4	2.3	4.2	4.9	6.0	-
Acetophenone	120	3.5	3.5	3.0	3.3	3.1	3.0	2.8	2.8
Nitrobenzene	123	1.5	1.6	1.6	1.6	1.6	1.7	1.8	-
Propiophenone	134	2.9	2.9	2.8	2.7	2.6	2.2	2.0	2.3
Acetophenone hydrazone	134	-	1.23	1.23	1.22	1.20	1.13	1.04	-
2-Methylbenzaldehyde hydrazone	134	-	0.63	0.63	0.59	0.62	0.60	0.53	0.63
4-Methylbenzaldehyde hydrazone	134	1.04	0.94	0.95	1.09	0.99	0.93	0.91	-
*p*-Toluilic acid	136	0.77	0.74	0.76	0.72	0.81	0.81	0.83	-
2-Hydroxybenzaldehyde hydrazone	136	0.48	0.47	0.45	0.46	0.41	0.40	0.40	-
3-Nitrophenol	139	0.39	0.33	0.38	0.40	0.41	0.45	0.45	-
Phthalimide	147	0.02	0.02	0.02	0.02	0.02	0.02	0.02	-
Butyrophenone	148	2.8	3.0	3.0	2.9	2.8	2.7	2.6	2.3
4-Methylacetophenone hydrazone	148	1.18	1.22	1.18	1.16	1.21	1.19	1.12	-
Propiophenone hydrazone	148	1.33	1.16	1.13	1.03	1.10	1.07	1.09	-
Butyrophenone hydrazone	162	1.30	1.13	1.12	1.08	1.10	1.07	1.01	-
1-Phenylpyrazolidin-3-one	162	2.6	2.7	1.3	1.6	2.2	1.7	2.3	-
Ninhydrine (hydrate)	178	-	0.42	0.48	0.46	0.47	0.44	0.44	-
Diethyl-*m*-toluamide	191	0.13	0.13	0.12	0.12	0.11	0.11	0.12	-
N-Allyl-*p*-toluenesulfonamide	211	0.063	0.062	0.061	0.059	0.059	0.058	0.057	0.052
N,N-Diethyl-*p*-toluenesulfonamide	227	0.28	0.28	0.28	0.27	0.21	0.26	0.24	0.26
N-*tert*-Butyl-*p*-toluenesulfonamide	227	0.091	0.091	0.087	0.092	0.092	0.093	0.094	0.095
N-Phenyl-*p*-toluenesulfonamide	247	0.26	0.26	0.26	0.26	0.25	0.29	0.27	0.25
N-Hexyl-*p*-toluenesulfonamide	255	0.07	0.069	0.067	0.066	0.066	0.064	0.062	0.064
N-Benzyl-*p*-toluenesulfonamide	261	0.078	0.075	0.074	0.072	0.074	0.065	0.068	0.066

**Table 5 molecules-28-00734-t005:** Relative optical densities *A*(254)/*A*(220) of some organic compounds, depending on the acetonitrile content in the eluent.

Analyte	MW	Acetonitrile Content (% *v*/*v*)						
		35	40	45	50	55	60	65
Toluene	92	0.133	0.133	0.128	0.131	0.132	0.134	0.135
*o*-Xylene	106	-	0.064	0.063	0.063	0.063	0.063	0.064
Chlorobenzene	112	-	-	0.057	0.063	0.070	0.074	0.078
1H-Benzotriazol	119	3.6	3.9	3.8	3.9	4.1	4.2	4.1
Acetophenone	120	4.1	3.8	3.4	3.4	3.0	2.9	2.7
Nitrobenzene	123	1.6	1.6	1.7	1.8	1.8	1.8	1.7
Propiophenone	134	3.2	3.0	2.7	2.6	2.4	2.2	2.2
2,3,5-Trimethylphenol	136	0.083	0.082	0.081	0.084	0.079	0.075	0.075
3-Nitrophenol	139	0.37	0.39	0.36	0.34	0.439	0.387	-
Phthalimide	147	0.031	0.033	0.035	0.037	0.039	0.041	0.042
Butyrophenone	148	2.4	2.3	2.2	2.1	2.0	1.9	1.9
1-Phenylpyrazolidin-3-one	162	1.39	1.40	1.43	1.44	1.45	-	-
Diethyl-*m*-toluamide	191	0.14	0.15	0.15	0.15	0.15	0.15	0.15
N-Allyl-*p*-toluenesulfonamide	211	0.076	0.073	0.071	0.074	0.067	0.074	0.065
N,N-Diethyl-*p*-toluenesulfonamide	227	-	0.295	0.293	0.288	0.290	0.290	0.290
N-*tert*-Butyl-*p*-toluenesulfonamide	227	0.113	0.110	0.108	0.107	0.104	0.102	0.106
N-Phenyl-*p*-toluenesulfonamide	247	0.31	0.29	0.29	0.28	0.28	0.30	0.30
N-Hexyl-*p*-toluenesulfonamide	255	-	-	0.077	0.074	0.073	0.072	0.071
N-Benzyl-*p*-toluenesulfonamide	261	-	0.16	0.16	0.15	0.16	0.15	0.15
Sulfamerazine	264	0.97	0.86	0.76	0.72	0.64	0.53	0.48

**Table 6 molecules-28-00734-t006:** Coefficients of the dependencies of the retention indices vs. the concentrations of the organic components in the eluents. All analytes are listed in the order of increasing *d*RI/*dC*(CH_3_OH) values.

Analyte	MW	*N*{H}	*d*RI/*dC*(CH_3_OH)	*d*RI/*dC*(CH_3_CN)	log*P* *
Compounds with *d*RI/*dC* << 0					
N-Hexyl-*p*-toluenesulfonamide	255	1	−5.6 ± 0.3	−3.1 ± 0.2	4.09 ± 0.30
N-Benzyl-*p*-toluenesulfonamide	261	1	−5.3 ± 0.2	−3.8 ± 0.1	3.21 ± 0.32
N-Phenyl-*p*-toluenesulfonamide	247	1	−5.2 ± 0.1	−4.5 ± 0.2	3.04 ± 0.29
N-*tert*-Butyl-*p*-toluenesulfonamide	227	1	−4.1 ± 0.2	−2.0 ± 0.1	2.66 ± 0.32
1-Phenylpyrazolidin-3-one	162	1	−4.0 ± 0.2	−2.5 ± 0.5	0.89
N-Allyl-*p*-toluenesulfonamide	211	1	−3.4 ± 0.1	−2.0 ± 0.1	2.26 ± 0.32
N,N-Diethyl-*p*-toluenesulfonamide	227	0	−3.2 ± 0.1	−1.8 ± 0.1	2.87 ± 0.28
Diethyl-*m*-toluamide	191	0	−2.0 ± 0.1	−1.2 ± 0.3	2.18
3-Nitrophenol	139	1	−1.0 ± 0.1	−1.9 ± 0.3	2.00
Compounds with *d*RI/*dC* ≈ 0					
Sulphamethoxazol (hydrate)	253	3	−0.4 ± 0.1	−1.4 ± 0.3	0.89
1H-Benzortriazole (probably, hydrate)	119	1	−0.24 ± 0.04	−2.2 ± 0.4	1.44
Phthalimide (probably, hydrate)	147	1	−0.24 ± 0.04	−1.5 ± 0.3	1.15
Acetophenone hydrazone	134	2	−0.11 ± 0.05	-	1.28 ± 0.51
Acetophenone	120	0	0.0	0.0	1.70
Ninhydrin (hydrate)	178	2	0.05 ± 0.08	-	0.67
Nitrobenzene	123	0	0.3 ± 0.2	0.5 ± 0.2	1.83
2,3,5-Trimethylphenol	136	1	0.3 ± 0.1	−1.5 ± 0.2	2.73
Compounds with *d*RI/*dC* >> 0					
Chlorobenzene	112	0	2.8 ± 0.2	0.8 ± 0.4	2.90
Toluene	92	0	4.0 ± 0.1	0.6 ± 0.2	2.71
*o*-Xylene	106	0	4.6 ± 0.3	0.2 ± 0.3	3.12

(*) Precalculated log*P* values are indicated with standard deviations (ACD software).

## Data Availability

All data will be included into PhD Theses of A. Derouiche and D. Nikitina and will be available after finalizing these works.

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
