# Peer review of "Evidence for the Hydration of Some Organic Compounds during Reverse-Phase HPLC Analysis"

_molecules, 2023, doi:10.3390/molecules28020734_

Round 1

Reviewer 1 Report

The paper is about the hydration of some organic compounds during reversed HPLC analysis. I think there is no problem with the paper to be published under MDPI molecules standard. I only give the following minor revision as follow:

Writing mistakes and error

1.       The font size for all of the equations are too large. Please reduce it and follow the guides from MDPI.

2.       The symbol of reaction direction in abstract line 8 is also too large. Please reduce it and change it to be similar with the abstract font size.

Author Response

Dear Reviewer,

Thank you for reviewing our manuscript. All mistakes and errors revealed have been corrected, namely:

- The font size of equations was reduced in accordance in accordance with Molecular Template;

- The symbol of reaction directions in the text was reduced, as well.

Besides that the font sized of all the parts of the text was brought into line with the Molecules Template.

On behalf of authors,

Igor G. Zenkevich

Reviewer 2 Report

General comments:

This manuscript introduced some methods to detect the formation of hydrates in water-containing eluent under the conditions of reversed phase HPLC analysis. The results of this study are meaningful and will get the interests of the researchers in the field of HPLC analysis. I think the paper is suitable for publication in Molecules. However, there are some minor problems need to be addressed prior to possible publication.

Specific Comments:

1. Figures: The definition of the figures is not enough and the resolution needs to be improved. Please show the captions of Y-coordinate at the corresponding locations.

2. I suggest that the author make the significance of the research more clearly in the introduction, including why the work was done and what scientific problem it was aimed to solve? 

3. Is it possible to correct the anomalies of retention times caused by the formation of hydrates?

4. The language of the manuscript should be carefully modified to make it simpler and easier to follow.

5. Line 81: Please provide the full course of LSM as it appears for the first time in the paper. 

6. Line 325: Please provide a formal table format.

Author Response

Dear Reviewer,

Thank you for reviewing our manuscript. Most of suggestions proposed have been taken into account, namely:

  1. The locations of the captions of Y-coordinates in Figures have been changed; all figures were re-drawn. No additional transformations with figures have been made to prevent decreasing their resolution;

  2. In the Introduction the statements of the problem considered have been updated as possible;

  3. It is possible to correct the anomalies caused by the formation of hydrates? The simplest way not to exclude these anomalies, but to minimize them is to replace acetonitrile in an eluent with methanol. This statement is included in the text (in the Conclusion);

  4. The English of the manuscript before its submitting has been corrected by the professional, but not native English speaking interpreter. I believe, that the subsequent improvement of English requires just native speaker;

  5. Thanks, corrected;

  6. The authors have tried to provide a formal Table format, but sometimes it seems difficult to change the form of data presentation.

Besides that the font sized of all the parts of the text was brought into line with the Molecules Template.

On behalf of authors,

Igor G. Zenkevich